# Mutant IDH1 Depletion Downregulates Integrins and Impairs Chondrosarcoma Growth

**DOI:** 10.3390/cancers12010141

**Published:** 2020-01-06

**Authors:** Luyuan Li, Xiaoyu Hu, Josiane E. Eid, Andrew E. Rosenberg, Breelyn A. Wilky, Yuguang Ban, Xiaodian Sun, Karina Galoian, Joanna DeSalvo, Jinbo Yue, Xi Steven Chen, Marzenna Blonska, Jonathan C. Trent

**Affiliations:** 1Sheila and David Fuente Graduate Program in Cancer Biology, University of Miami Miller School of Medicine, Miami, FL 33136, USA; l.li22@med.miami.edu; 2Department of Medicine, Division of Hematology and Oncology, University of Miami Miller School of Medicine, Miami, FL 33136, USA; xiaoyuhu0523@foxmail.com (X.H.); jee64@med.miami.edu (J.E.E.); breelyn.wilky@cuanschutz.edu (B.A.W.); jdesalvo@med.miami.edu (J.D.); mblonska@med.miami.edu (M.B.); 3Sylvester Comprehensive Cancer Center, University of Miami Miller School of Medicine, 1120 NW 14th Street, Miami, FL 33136, USA; arosenberg@med.miami.edu (A.E.R.); yuguang.ban@med.miami.edu (Y.B.); xiaodian.sun@med.miami.edu (X.S.); steven.chen@med.miami.edu (X.S.C.); 4School of Medicine and Life Sciences, University of Jinan Shandong Academy of Medical Sciences, Jinan 250000, Shandong, China; yuejinbo@hotmail.com; 5Department of Radiation Oncology, Shandong Cancer Hospital and Institute, Shandong First Medical University and Shandong Academy of Medical Sciences, Jinan 250000, Shandong, China; 6Department of Pathology, University of Miami Miller School of Medicine, Miami, FL 33136, USA; 7Department of Medicine, University of Colorado School of Medicine, Aurora, CO 80045, USA; 8Department of Orthopaedic Surgery, University of Miami Miller School of Medicine, Miami, FL 33136, USA; kgaloian@med.miami.edu; 9Department of Public Health Sciences, University of Miami Miller School of Medicine, Miami, FL 33136, USA

**Keywords:** chondrosarcoma, IDH mutation, integrin, 2-hydroxyglutarate, CRISPR/Cas9

## Abstract

Chondrosarcomas are a heterogeneous group of malignant bone tumors that produce hyaline cartilaginous matrix. Mutations in isocitrate dehydrogenase enzymes (IDH1/2) were recently described in several cancers, including conventional and dedifferentiated chondrosarcomas. These mutations lead to the inability of IDH to convert isocitrate into α-ketoglutarate (α-KG). Instead, α-KG is reduced into D-2-hydroxyglutarate (D-2HG), an oncometabolite. IDH mutations and D-2HG are thought to contribute to tumorigenesis due to the role of D-2HG as a competitive inhibitor of α-KG-dependent dioxygenases. However, the function of IDH mutations in chondrosarcomas has not been clearly defined. In this study, we knocked out mutant IDH1 (IDH1^mut^) in two chondrosarcoma cell lines using the CRISPR/Cas9 system. We observed that D-2HG production, anchorage-independent growth, and cell migration were significantly suppressed in the IDH1^mut^ knockout cells. Loss of IDH1^mut^ also led to a marked attenuation of chondrosarcoma formation and D-2HG production in a xenograft model. In addition, RNA-Seq analysis of IDH1^mut^ knockout cells revealed downregulation of several integrin genes, including those of integrin alpha 5 (ITGA5) and integrin beta 5 (ITGB5). We further demonstrated that deregulation of integrin-mediated processes contributed to the tumorigenicity of IDH1-mutant chondrosarcoma cells. Our findings showed that IDH1^mut^ knockout abrogates chondrosarcoma genesis through modulation of integrins. This suggests that integrin molecules are appealing candidates for combinatorial regimens with IDH1^mut^ inhibitors for chondrosarcomas that harbor this mutation.

## 1. Introduction

Chondrosarcoma constitutes a heterogeneous group of malignant bone tumors that produce hyaline cartilaginous matrix. It is the second most common primary malignancy of bone following osteosarcoma. Chondrosarcomas that arise de novo are called primary chondrosarcomas while those superimposed on preexisting benign cartilaginous neoplasms such as enchondroma or osteochondroma are referred to as secondary chondrosarcomas [1]. Approximately 85% of all chondrosarcomas are the conventional subtypes further classified into central, peripheral and periosteal lesions according to the osseous location. The remaining 10–15% consist of rare subtypes including dedifferentiated, mesenchymal, and clear cell chondrosarcoma. Conventional chondrosarcoma most often arises in individuals over 40 years of age, and may be associated with significant morbidity and mortality [2,3]. Poor patient outcomes may be related to primary resistance to conventional chemotherapy and radiotherapy, leaving surgery as the only effective treatment option. Chondrosarcomas tend to recur following initial resection and many patients with high grade tumors eventually develop metastatic disease. Recurrence may be associated with biological progression as greater than 10% of recurrent chondrosarcomas are of higher histological grade than the original neoplasm [1]. In the absence of effective systemic therapy, metastatic disease is nearly uniformly fatal. Therefore, better therapy for patients with unresectable or metastatic disease is urgently needed.

Isocitrate dehydrogenases (IDHs) are enzymes that convert isocitrate to α-ketoglutarate (α-KG) with production of NADH or NADPH, and play key roles in cellular metabolism. IDHs occur as three distinct subtypes: NADP^+^-dependent IDH1 and IDH2 and NAD^+^-dependent IDH3. IDH1 is localized in the cytoplasm and peroxisomes whereas IDH2 and IDH3 are mitochondrial [4]. IDH1/2 mutations have been found in gliomas, hematological malignancies, and other tumors such as cholangiocarcinoma, melanoma, prostate cancer, and thyroid carcinoma [5]. Recently, IDH1/2 mutations have been reported in enchondroma, as well as in conventional (central and periosteal) and dedifferentiated chondrosarcoma [5,6]. They represent a novel genetic abnormality in chondrosarcoma, indicating a potential role for aberrant IDH function in the pathogenesis of this malignancy. IDH mutations lead to the inability of IDH to convert isocitrate into α-KG. Instead, α-KG is reduced to an oncometabolite, D-2-hydroxyglutarate (D-2HG) [7]. In gliomas and leukemias, IDH mutations have been shown to initiate tumorigenesis by D-2HG acting as a competitive inhibitor of α-KG-dependent dioxygenases such as the TET family of 5-methylcytosine hydroxylases, JumonjiC domain-containing histone demethylases (JHDMs) and the prolyl hydroxylases (PHDs) [5,8]. Nonetheless, the role of IDH mutation in the malignant transformation and development of chondrosarcomas remains unclear.

The anti-tumor activity of an IDH1^mut^ inhibitor in chondrosarcoma cells was previously reported [9]. Herein, we sought to analyze the role of IDH1 mutation in chondrosarcoma tumorigenesis by knockout of the IDH1^mut^. Using the CRISPR/Cas9 technique, we established single cell-derived clones with inactive IDH1^mut^ in two chondrosarcoma cell lines. We found that loss of IDH1^mut^ attenuated the tumorigenicity of chondrosarcoma cells and led to a marked attenuation of chondrosarcoma formation and D-2HG production. To understand the underlying mechanism, we performed RNA-Seq (RNA Sequencing) analysis in the IDH1^mut^ knockout cells which revealed a significant downregulation of integrin genes such as integrin alpha 5 (ITGA5) and beta 5 (ITGB5). In vitro functional assays further demonstrated that integrin-mediated cell-matrix interactions contributed to the tumorigenicity of IDH1-mutant chondrosarcoma cells. Our results highlight a role of mutant IDH1 in chondrosarcoma genesis through deregulation of integrin function, and present integrins as potential therapeutic targets for the treatment of IDH-mutant chondrosarcomas.

## 2. Materials and Methods

### 2.1. Cell Culture

The JJ012 cell line was provided by Dr. Joel Block and the HT1080 cell line was purchased from ATCC (Manassas, VA, USA). JJ012 and HT1080 were grown in RPMI-1640 medium supplemented with 10% fetal bovine serum (FBS) and 1% penicillin/streptomycin. The C28 cell line was provided by Dr. Karina Galoian and grown in 1:1 DMEM/F12 medium supplemented with 10% FBS and 1% penicillin/streptomycin.

### 2.2. IDH1 Knockout by CRISPR/Cas9 Technology

The CRISPR/Cas9 plasmid products and support reagents including IDH1 CRISPR/Cas9 Knockout (KO) Plasmids, Homology-Directed Repair (HDR) Plasmids, Ultracruz transfection reagent, plasmid transfection medium and puromycin dihydrochloride were purchased from Santa Cruz Biotechnology (Dallas, TX, USA). 1.5 × 10^5^ cells/well in 3 mL of antibiotic-free standard growth medium were seeded in 6-well plates, 24h prior to transfection. For each transfection, 2 µg of IDH1 KO plasmids and HDR plasmids were diluted in plasmid transfection medium and mixed with transfection reagent. After a 30 min incubation at room temperature, the mixture was added dropwise to each well. Expression of KO and/or HDR plasmids was visually confirmed by detection of GFP and/or RFP via fluorescent microscopy, 72 h post-transfection. Cells were further selected with 1 µg/mL puromycin for 5 days. After selection, cells with RFP were sorted by a FACS Aria Fusion cell sorter (BD Biosciences, San Jose, CA, USA) and seeded as single cells into 96-well plates. Single-cell colonies were isolated 10 days later followed by numerical expansion and analysis.

### 2.3. Polymerase Chain Reaction (PCR), Reverse Transcriptase PCR (RT-PCR) and Quantitative RT-PCR (qRT-PCR)

For PCR, genomic DNA was isolated using DNeasy Blood & Tissue Kit (Qiagen, Germantown, MD, USA). Sense primer-GACTTCCTTCCATGGTCACCTC and antisense primer-GTGCCCT CCATGTACAGCTTCA were used for JJ012. Sense primer CAGGAAGGCAGAGGTTGAGTGA and antisense primer GTGCCCTCCATGTACAGCTTCA were used for HT1080. For RT-PCR and qRT-PCR, total RNA was isolated using miRNeasy Mini Kit (Qiagen) and cDNA was synthesized from 1 µg of total RNA using iScript cDNA synthesis kit (Bio-Rad, Hercules, CA, USA) in a 20 µL total volume. Sense primer GTGGCGGTTCTGTGGTAGAGAT and anti-sense primer GCCAACCCTTAGACAGAGCCAT were used in RT-PCR. The PCR products were purified using the QIAquick PCR purification kit (Qiagen) and sequenced using the same primers (Source Bioscience, Nottingham, UK). The sequencing data were analyzed using Finch TV software. qRT-PCR was set up with 2 µL of 1:5 diluted cDNA, 20x TaqMan probes and TaqMan universal PCR Master Mix (ThermoFisher Scientific, Waltham, MA, USA) in 20 µL total volume. Samples were run in triplicate on a BioRad CFX-96 real time system. Gene expression levels were calculated using the 2^−ΔΔCt^ method [10]. The following gene-specific TaqMan primers/probe sets were used: GAPDH (internal normalization control), ITGA5, ITGA10, ITGA2, ITGB2 and ITGB5.

### 2.4. Western Blotting

Cells were directly lysed by the addition of Laemmli Sample Buffer (BioRad) supplemented with 2-mercaptoethanol (Sigma-Aldrich St. Louis, MO, USA). Protein lysates from tumors were obtained using RIPA buffer (Thermo Scientific, Waltham, MA, USA), supplemented with protease inhibitor cocktail (Sigma), phosphatase inhibitor cocktail 2 (Sigma) and phosphatase inhibitor cocktail 3 (Sigma) at a 1:100 ratio. Briefly, 100 µg of tumor tissue was homogenized in 500 µL of RIPA buffer using a homogenizer and TissueRuptor disposable probes (Qiagen). The homogenates were centrifuged at 14,000 rpm for 10 min at 4 °C, and the supernatants were collected. Cell and tissue lysates were denatured at 95 °C for 10 min with Laemmli Sample Buffer. Equal total protein was electrophoretically fractionated in 4–20% Mini-PROTEIN TGX precast gels (Bio-Rad), transferred onto nitrocellulose membrane (Pall Corporation, New York, NY, USA) and subjected to immunoblot analysis with specific antibodies against IDH1 (1:1000, Abcam, Cambridge, MA, USA), IDH2 (1:1000, Abcam, Cambridge, MA, USA), IDH3A (1:1000, Abcam, Cambridge, MA, USA), ITGA5 (1:1000, Abcam, Cambridge, MA, USA), ITGB5 (1:1000, Abcam, Cambridge, MA, USA), ITGA10 (1:2000, Millipore Sigma, Burlington, MA, USA), FAK (1:1000, Cell Signaling, Danvers, MA, USA), Phospho-FAK (Tyr397) (1:1000, Cell Signaling) and β-actin (1:5000, Cell Signaling). Autoradiography of the membranes was performed using the ChemiDoc MP Imaging System (Bio-Rad) with West Femto Maximum Sensitivity Substrate (Thermo Scientific). Densitometry of images was carried out via the ImageJ Analyze Gels (NIH, Bethesda, MD, USA) module and normalized to the corresponding β-actin signal for each band.

### 2.5. Cell proliferation, Colony Formation, Transwell Migration and Cell Adhesion Assays

Cell proliferation was assessed by CellTiter 96 AQueous One Solution Cell Proliferation Assay (Promega Corporation, Madison, WI, USA). 2 × 10^3^ cells per well were plated in a 96-well plate and cultured for 1 to 5 days. Assays were performed by adding 20 µL of the reagent directly to culture wells, incubating for 1h, and recording the absorbance at 490 nm with a plate reader (BioRad iMaxi). The colony formation assay was performed according to the protocol described by Borowicz et al. [11]. Briefly, 5 × 10^3^ cells per well were incubated in an upper layer of 0.3% agar in RPMI 1640 with 10% FBS overlaid on a lower layer of 0.5% basal agar. Cells were maintained in a 5% CO_2_ incubator at 37 °C for 10–14 days (JJ012 and HT1080 cells) or 4 weeks (C28 cells). Colonies were stained with 2 mg/mL iodonitrotetrazolium chloride (Sigma-Aldrich) overnight and counted using GelCount software (Oxford Optronix, Abingdon, UK). Cell migration assays were carried out using 8-µm transwell inserts (Corning Life Sciences, Corning, NY, USA). Briefly, 0.5 mL of cell suspension containing 2.5 × 10^4^ cells or 2 mL containing 2.5 × 10^5^ cells in serum-free culture medium was loaded into the upper chambers of a 24-well or 6-well insert, respectively. For the integrin blockade assay, JJ012 cells were pretreated with 1 μg/mL ITGα5β1-blocking antibody (Novus, Centennial, CO, USA) or IgG isotype control (Novus), and HT1080 cells were pretreated with 10 μg/mL ITGαvβ5-blocking antibody (Novus) or IgG isotype control (Novus) for 2h at room temperature. The lower chambers were filled with 0.75 mL (24-well plate) or 2.5 mL (6-well plate) of complete media containing 10% FBS. After a 22h incubation, the non-invading cells were removed from the upper surface of the membrane with cotton swabs. The cells on the lower surface of the membrane were fixed and stained using the Differential Quick Staining Kit (Electron Microscopy Sciences, Hatfield, PA, USA). Cells were then counted manually after photographing the membrane through the microscope. For the adhesion assay, cells were detached using TrypLE buffer (ThermoFisher Scientific, Waltham, MA, USA). 2 × 10^4^ cells from each group were layered on fibronectin-coated 6-well plates that were pretreated with PBS containing 1% heat-inactivated BSA. For the adhesion blockade assay, JJ012 cells were pretreated with 1 μg/mL ITGα5β1-blocking antibody (Novus) or IgG isotype control antibody (Novus) and HT1080 cells were pretreated with 10 μg/mL ITGαvβ5-blocking antibody (Novus) or IgG isotype control antibody (Novus) for 2 h at room temperature. Cells were allowed to adhere for 1–2 h at 37 °C in a 5% CO_2_ incubator followed by washing, fixing, and staining. Cell adhesion was evaluated by counting the average number of attached cells per field. All assays were set up with 3 or more replicates per condition, and at least 3 independent experiments were carried out.

### 2.6. Measurement of D-2HG and α-KG

Measurement of D-2HG and α-KG was performed at MtoZ Biolabs (Boston, MA, USA). Metabolites were extracted from 1 × 10^7^ cells or 20 mg homogenized tumor tissue. Quantitative analyses of D-2HG and α-KG were conducted by high-performance liquid chromatography–tandem mass spectrometry (HPLC-MS/MS) with single-reaction monitoring (SRM) scans according to the protocol described by Cheng et al. and Han et al., respectively [12,13].

### 2.7. Xenograft Chondrosarcoma Model

All animal experiments were performed in compliance with University of Miami Institutional Animal Care and Use Committee (IACUC)-approved protocol (No. 19-079). Briefly, 2 × 10^6^ cells were resuspended in serum-free RPMI-1640 media and mixed with Matrigel at a 1:1 ratio. 200 µL of mixture was injected subcutaneously using a 26-gauge needle into the right flank of 4–6-week-old female nude mice (*n* = 8) (Nu/Nu, Jackson Labs, Bar Harbor, ME, USA). Tumor volumes were measured with electronic precision calipers (VWR, Radnor, PA, USA) according to the formula *V* = 0.5 × L (*length*) × W^2^ (*width*). Tumor-bearing mice were euthanized when tumors in any group exceeded 10% of animal body weight (~2 cm^3^). Immediately following euthanasia, tumors were excised, weighed, photographed, and sectioned into samples for formalin (10%) fixation or snap-frozen in liquid nitrogen.

### 2.8. RNA-Seq and Ingenuity Pathway Analysis (IPA)

Total RNA was extracted using miRNeasy Mini Kit (Qiagen). RNA-Seq library preparation was conducted using Illumina TruSeq Total Stranded RNA with RiboZero Gold (Illumina Inc., San Diego, CA, USA) and paired-end 75 bp sequencing was conducted using an Illumina NextSeq 500 platform to generate 30–39 million read pairs per sample. RNA-Seq analysis was performed to identify differentially expressed genes (DEGs) between the KO cells and the parental cells. Specifically, raw reads were aligned to human genome (GRCh38) using STAR (ver. 2.3.0e) [14]. Differential expression was analyzed using DESeq2 (ver. 1.20.0) [15], and false discovery rate (FDR)-adjusted *p* value < 0.05 were considered statistically significant. Heat maps were generated on normalized expression with hierarchical clustering. Pathway analysis was performed using IPA (Qiagen, www.qiagen.com/ingenuity).

### 2.9. Flow Cytometry

The cell surface expression of ITGα5β1 and αvβ5 was determined by flow cytometry. 1 × 10^6^ cells were harvested using TrypLE. After washing, JJ012 cells were stained with 5 µL of Alexa Fluor 488-labeled anti-ITGα5β1 antibody (volociximab, Novus Biologicals, Centennial, CO, USA) or IgG isotype control (Novus Biologicals, Centennial, CO, USA) in 45 µL Flow Cytometry Staining Buffer (ThermoFisher Scientific, Waltham, MA, USA) for 30 min on ice in the dark. HT1080 cells were stained with 5 µL of BV421-labeled anti-ITGαvβ5 antibody (BD Biosciences, San Jose, CA, USA) or IgG2bκ isotype control (BD Biosciences, San Jose, CA, USA) under the same conditions. After a single wash with staining buffer, the cells were fixed in 4% paraformaldehyde (ThermoFisher Scientific, Waltham, MA, USA) and read on an LSR-II analyzer (BD Biosciences, San Jose, CA, USA).

### 2.10. Stable Expression of IDH1^wt^

The lentivirus vector pLV[Exp]-EGFP/Neo-EF1A>hIDH1[NM_005896.3]*/3xFLAG (ID: VB171011-1031umv) was used to overexpress Flag-tagged full length IDH1^wt^. A blank vector pLV[Exp]-EGFP:T2A:Neo-Null (ID: VB160420-1010zqm) was used as a negative control. Both vectors were constructed and packaged by VectorBuilder (Cyagen Biosciences, Santa Clara, CA, USA) and the detailed information can be retrieved on www.vectorbuilder.com. Chondrosarcoma cells were infected with 5 multiplicity of infection (MOI), and 5 µg/mL polybrene was added to the cultures. After overnight culturing medium was changed, cells were split 48h later, and grown thereafter in 600 µg/mL geneticin (G418) for selection of infected cells.

### 2.11. Statistical analysis

Statistical analyses were conducted using GraphPad Prism 7 software (GraphPad Software, San Diego, CA, USA). Differences between two groups were analyzed using an unpaired two-tailed *t* test and *p* < 0.05 was considered statistically significant.

## 3. Results

### 3.1. Knockout of IDH1^mut^ in Two Human Chondrosarcoma Cell Lines

We have previously reported that pharmacological inhibition of IDH1^mut^ in human chondrosarcoma cells led to decreased production of D-2HG and suppressed their tumorigenic properties [9].

This provided the first evidence that IDH mutation is associated with chondrosarcoma tumorigenesis, but its mechanistic function has not been clearly defined. To further determine the role of IDH mutation in chondrosarcomas, we aimed to establish a cell model with total inactivation of IDH1^mut^ and depletion of D-2HG production. We used two cell lines, JJ012 and HT1080, both of which have been reported to harbor a heterozygous IDH1 point mutation [9,16]. To be noted, HT1080 was originally reported as a fibrosarcoma of bone, but this cell line is now considered to represent a dedifferentiated chondrosarcoma due to the presence of IDH1 mutations [17,18]. Knockout of IDH1^mut^ was achieved by transduction of IDH1 CRISPR/Cas9 KO plasmids which induce a site-specific double strand break (DSB) at the *IDH1* loci. Repair of the DSB by the IDH1 HDR plasmids incorporates RFP and puromycin resistance cassettes that allow the selection of transfected cells. The application of the CRISPR/Cas9 system is described in Figure 1A.

Single-cell colonies of stable JJ012 and HT1080 transfectants were successfully isolated, and among them, we identified one mock control clone with intact IDH1^wt^/IDH1^mut^ and two KO clones with disrupted IDH1^wt^/IDH1^mut^ for each cell line (Figure 1B). Loss of IDH1 was verified at the DNA [Figure 1B, upper panel], RNA [Figure 1B, middle panel], and protein levels (Figure 1B, lower panel). We describe the mock control clone as a stably transfected cell line that failed to induce a DSB in the *IDH1* loci, and we therefore decided to utilize it as a control for non-specific effects in our assays. Analysis of transcripts derived from the disrupted IDH1 locus by RT-PCR in the two HT1080 KO clones [Figure 1B, middle panel, right] revealed a deletion of 177 bp coding sequence containing the mutation site (Figure 1C). However, this deletion was not detected in the JJ012 KO cells, indicating that the genetic disruption of these cells was different from that of HT1080 cells. Notably, D-2HG production was almost completely suppressed in IDH1^mut^ KO clones derived from both cell lines (Figure 1D).

It should be noted, since all presently identified hotspot mutations are single-nucleotide substitutions in the respective arginine codons, the CRISPR/Cas9 KO plasmids are not able to differentiate IDH1^wt^ from IDH1^mut^, and thus tend to cause a homozygous knockout. Indeed, western blot analysis of all the generated single cell-derived KO clones showed a complete loss of both IDH1^wt^ and IDH1^mut^ expression. Interestingly, the results of our model are similar to the effects of specific IDH1^mut^ inhibitors such as FDA-approved AG-120 and AG-221, which have been shown to inhibit IDH^wt^ as well [19]. Since IDH^wt^ produces α-KG which is then used by IDH^mut^ to produce D-2HG, it is conceivable that the attenuated IDH^wt^ activity may render targeting IDH^mut^ more effective and could partially contribute to any observed anti-tumor activity. However, in our study, the α-KG levels remained unchanged upon IDH1^mut^ knockout (Appendix A), rendering this process an unlikely scenario. Moreover, to confirm the specificity of the IDH1 knockout, we tested IDH2 and IDH3 levels in cells and found their expression unchanged in the KO clones of both cell lines (Appendix A). Regardless, our cell line panel provides a unique model to functionally study the role of IDH1 mutation in chondrosarcoma.

### 3.2. Loss of IDH1^mut^ Attenuated the Tumorigenicity of Chondrosarcoma Cells

To study the function of IDH1 mutation in chondrosarcoma, we examined the effects of IDH1^mut^ knockout on cell proliferation, cell migration, and anchorage-independent cell growth, in the genetically modified JJ012 and HT1080 cells. Consistent with previous results [9], loss of IDH1^mut^ in the KO chondrosarcoma cell lines failed to induce significant changes in cell proliferation (Figure 2A), supporting the idea that IDH1 mutations are dispensable for chondrosarcoma cell proliferation. Moreover, anchorage-independent cell growth is the ability of transformed cells to grow independently of a solid surface. It is a fundamental property of cancer cells and tightly correlates with tumorigenic and metastatic potential in vivo [20]. Strikingly, we found that depletion of IDH1^mut^ led to a marked reduction in the capacity of the JJ012 and HT1080 cells for anchorage-independent growth in soft agar (Figure 2B). This result suggests a critical role of IDH1^mut^ in the tumorigenic potential of chondrosarcoma cells [20]. Furthermore, cell migration is a component of cell movement in vitro and known to contribute to cancer metastasis [21]. We observed that knockout of IDH1^mut^ in the chondrosarcoma cells significantly decreased the number of migratory cells in both lines (Figure 2C). These findings are consistent with our previous data that showed impaired chondrosarcoma cell migration upon treatment with an IDH1^mut^ inhibitor [9] and implicate IDH1^mut^ in mediating chondrosarcoma cell migration.

### 3.3. The Attenuated Tumorigenicity of Chondrosarcoma Cells Was Not Caused by IDH1^wt^ Loss

As described before, both IDH1^wt^ and IDH1^mut^ were knocked out in our cell model. To further confirm that it is the loss of IDH1^mut^ rather than that of IDH1^wt^ is responsible for the attenuated tumorigenicity of chondrosarcoma cells, we restored wild-type IDH1 expression in JJ012 and HT1080 IDH1 KO cells through the introduction of lentiviral vectors expressing Flag-tagged IDH1^wt^. EGFP expression was detected in the majority of cells infected with the blank (titer of 2.16 × 10^9^ TU/mL; MOI of 5) or recombinant IDH1^wt^ vectors (titer of 8.15 × 10^8^ TU/mL; MOI of 5), reflecting the high efficiency of lentiviral infections in the IDH1 KO cells from both cell lines (Appendix A). Immunoblots confirmed high levels of exogenous IDH1 expression in the IDH1^wt^-transduced IDH1 KO cells (Figure 3A). As expected, HPLC-MS analysis of these cells revealed a slight increase in α-KG levels (Appendix A) but unchanged D-2HG levels (Appendix A). Importantly, re-expression of IDH1^wt^ in the IDH1 KO cells did not alter their phenotypes of decreased colony formation in soft-agar (Figure 3B) and cell migration (Figure 3C). For an additional control, we employed C28, an immortalized human chondrocyte cell line that expresses IDH1^wt^ only [9]. Knockout of IDH1 in C28 cells was achieved using the same CRISPR/Cas9 system, and a pool of IDH1 knockout cells were used for functional assays. As expected, knockout of IDH1 did not affect colony formation in soft-agar or cell migration of C28 cells (Appendix A). Altogether, our findings in both experiments clearly demonstrate that the attenuated tumorigenicity of chondrosarcoma cells was caused by loss of the IDH1^mut^ rather than of the IDH1^wt^ allele.

### 3.4. Loss of IDH1^mut^ Led to Suppression of Chondrosarcoma Growth and D-2HG Production

The effects of IDH1^mut^ knockout on anchorage-independent growth and migration led us to test whether IDH1 mutation confers a tumor growth advantage in vivo We subcutaneously implanted 2 × 10^6^ cells from two KO clones, a mock control clone and a parental control from each cell line into nude mice (*n* = 8 per group). Tumor incidence and growth rates within each group were closely monitored. 5 mice from each of the JJ012-derived groups developed tumors 3 weeks post-implantation. Importantly, the tumors in the KO groups grew at a significantly slower rate and measured 50% or less in volume compared to those in the control groups (mock and parental) (Figure 4A, left). The HT1080 cells are capable of forming very aggressive tumors. All 16 mice from the two control groups and 11 of 16 mice from the KO groups developed tumors within 1-week post-implantation. In 2 weeks, the majority of tumors in the control groups were above 1000 mm^3^ in volume while over half of the tumors in the KO groups were under 500 mm^3^ in volume (Figure 4A, right).

The mean tumor weight in the KO groups determined at the endpoint was approximately 30% of those in the control groups from both cell lines (*p* < 0.05) (Figure 4B). The chondrosarcoma tumors were then excised and analyzed for IDH1 expression. Immunoblotting revealed complete suppression of IDH1 protein in all tumors derived from the KO groups (Figure 4C). These results implicate IDH1^mut^ in chondrosarcoma tumor growth.

High levels of D-2HG produced by IDH1^mut^ are closely associated with tumorigenesis. D-2HG tends to accumulate at high levels in IDH-mutant tumors. Choi et al. reported that D-2HG levels in IDH1- and IDH2-mutated tumors were 20-fold to 2000-fold higher than those in wild-type IDH glioblastomas [22]. We then tested whether loss of IDH1^mut^ led to a suppression of D-2HG production in the xenograft models. We found that D-2HG levels in all the tumors from the KO groups were reduced by approximately 50-fold compared to D-2HG levels in the control groups (Figure 4D).

### 3.5. Loss of IDH1^mut^ Downregulates Integrins

To understand the molecular mechanism of the anti-tumor effects due to IDH1^mut^ knockout, we performed RNA-Seq in triplicate on two IDH1^mut^ KO clones and a parental control of each chondrosarcoma cell line. RNA-Seq analysis of the JJ012 cells identified 1104 differentially expressed genes (DEGs) common to its two KO clones as compared with the parental control (FDR-adjusted *p* value < 0.05) (Figure 5A, left). Of these, 506 were up-regulated and 598 were down-regulated. In the HT1080 cells, 1518 DEGs common to its two KO clones as compared with the parental control were identified (FDR-adjusted *p* value < 0.05) (Figure 5A, right). Of these, 863 were up-regulated and 655 were down-regulated. Functional categorization revealed that the affected genes were mediators of cell movement, cell death and survival, cell-to-cell interactions, and others (Figure 5B, left). As anticipated from the impaired cell migration of the IDH1^mut^ knockouts, genes belonging to adhesion/integrin- related pathways such as integrin signaling, focal adhesion kinase (FAK), integrin-linked kinase (ILK), and actin cytoskeleton signaling were highly represented in the transcriptome of the IDH1^mut^ KO cells (Figure 5B, right). Notably, a large subset of the integrin family (30–50%) were among the most highly regulated genes in the IDH1^mut^ KO clones of both cell lines (Figure 5C). Validation of the RNA-Seq results by qRT-PCR revealed significant downregulation of ITGA10, ITGA5, and ITGA2 integrin genes in JJ012 KO cells, and ITGA10, ITGB5, and ITGB2 integrin genes in HT1080 KO cells (Figure 5D). Of the validated integrins, ITGA5 and ITGB5 were the most downregulated at the protein level, in the IDH1^mut^ KO cells (Figure 5E) and tumors (Figure 5F). Altogether, these findings clearly suggest an association between IDH mutation and an aberrant activation of integrin signaling in chondrosarcoma cells. We then asked whether the aberrant activation of integrin signaling in chondrosarcoma cells is associated with production of the oncometabolite, D-2HG. Integrin activation induces the autophosphorylation of FAK at tyrosine 397, which initiates the outside-in signaling cascade [23]. Activation of the FAK complex is central to the regulation of downstream signaling pathways that control cell spreading, cell movement and cell survival [24]. We found that FAK phosphorylation at tyrosine 397 was decreased in both JJ012 and HT1080 IDH1 KO cells compared with their parental controls. Notably, treatment with a cell permeable form of D-2HG, octyl-D-2HG, led to a significant increase of FAK phosphorylation in these KO cells [Appendix A]. These results suggest a cause-to-effect link between IDH mutation, D-2HG production and FAK activation downstream of integrin signaling. Furthermore, in vitro and in vivo data have implicated a number of integrins including integrin α5 and β5 in the regulation of cell growth, survival and migration during angiogenesis [25]. Thus, we hypothesized that IDH mutation/D-2HG linked integrin signaling is a promoter of angiogenesis. To test this hypothesis, we began by examining the expression of an endothelial marker, ERG in JJ012 cell-derived tumors by immunohistochemistry. We found that ERG expression was significantly decreased in the IDH1^mut^ KO tumors [Appendix A], suggesting a functional association of IDH mutation with angiogenesis.

### 3.6. Integrin Activation Contributes to the Tumorigenicity of Chondrosarcoma Cells

Integrins are heterodimeric transmembrane glycoproteins which are composed of an α and a β subunit. The local expression pattern of integrins and their extracellular matrix (ECM) ligands controls the response of cells to their microenvironment, as each individual integrin heterodimer is capable of binding multiple ligands, and each ligand can bind multiple integrin dimers [26]. Integrin-mediated interactions between the cancer cell and ECM are essential for cell growth, migration, and survival in the process of tumor development and spread, including chondrosarcomas [27,28,29]. Considering the broad downregulation of integrins in IDH1^mut^ KO clones, we hypothesized that the loss of integrin expression in these cells may cause an interruption of cell-ECM interaction that would lead to a reduction in their tumorigenic properties in vitro and in vivo. Integrins α5 (ITGA5) and β5 (ITGB5) were the most abundant and downregulated integrins upon knockout of IDH1^mut^ in JJ012 and HT1080, respectively. Studies have indicated that integrin α5 expression correlates with cancer progression and plays an important role in enhancing cancer cell adhesion and migration through a fibronectin matrix [30,31,32]. Integrin β5 has been reported to facilitate tumorigenicity through promotion of cancer cell migration, anchorage-independent growth, angiogenesis, and epithelial-mesenchymal transition (EMT) [33,34]. Thus, we further pursued the specific functions of these two molecules in the IDH-mutant chondrosarcoma cells.

The ITGα5β1 heterodimer preferentially binds fibronectin while ITGαvβ5 heterodimer preferentially binds vitronectin [35]. To assess whether formation of the ITGα5β1 and ITGαvβ5 complexes in the chondrosarcoma cells are associated with IDH mutation, we carried out flow cytometry assays with antibodies recognizing surface epitopes of these complexes. First, we validated the expression of these two heterodimers in both parental cell lines (Figure 6A). Consistent with the RNA-Seq results, cells depleted of IDH1^mut^ displayed 20–40% less ITGα5β1 and ITGαvβ5 heterodimers compared with control cells as indicated by the median fluorescence intensity (MFI) (Figure 6B). We then examined the contribution of ITGα5β1 and ITGαvβ5 to cell–matrix interactions in adhesion and migration assays. We found that IDH1^mut^ knockout in JJ012 cells severely reduced their adhesion to fibronectin; the number of cells attached to fibronectin in the IDH1^mut^ KO cells was about 30% of the number in the control cells (*p* < 0.05) (Figure 6C). Interestingly, blockade of ITGα5β1 in JJ012 cells using a neutralizing antibody abolished their adhesion ability (*p* < 0.01) (Figure 6D). These results indicate that IDH1^mut^ is required for effective adhesion of JJ012 cells to fibronectin, and loss of ITGα5β1 contributes to the reduced adhesion capacity of the IDH1^mut^ KO clones. It should be noted that adhesion to vitronectin was not altered in the HT1080 IDH1^mut^ KO cells, or in HT1080 cells pretreated with neutralizing ITGαvβ5 antibody [Appendix A]. However, blockade of ITGα5β1 and ITGαvβ5 dramatically decreased migration of JJ012 and HT1080 cells, respectively (Figure 6E), demonstrating that loss of these two integrin heterodimers contributes to the observed reduced migration in the IDH1^mut^ KO clones. Overall, our data showed that integrin-mediated cell-ECM interactions contribute to IDH1-mutant chondrosarcoma cell migration and/or cell adhesion.

## 4. Discussion

Locally advanced or metastatic chondrosarcoma is common and particularly problematic due to limited treatment options. Development of novel treatments is critical to patients with tumors that are not amenable to surgical resection. The discovery of IDH mutation in chondrosarcoma identified a potential therapeutic target that may provide a novel treatment strategy and improve outcomes for patients. However, elucidation of the role of IDH mutation in chondrosarcoma has been hampered by a lack of appropriate cell-based models. Previous studies using IDH^mut^ inhibitors or shRNA systems have the limitation of incomplete depletion of D-2HG and possible off-target effects associated with the majority of pharmacological agents. Moreover, expression of IDH^mut^ in established cancer cell lines does not necessarily recapitulate the oncogenesis process caused by IDH mutation in the human tumors, as these cells may be transformed through a pathway independent of IDH mutations and thus pathologically irrelevant. Instead, in this study, we investigated the role of IDH mutation in chondrosarcoma by directly knocking out the mutant allele in two IDH1-mutant chondrosarcoma cell lines, JJ012 and HT1080. To avoid the potential non-specific and off-target effects for each cell line, we identified one mock control clone with intact IDH1^wt^/IDH1^mut^ which serves as an additional control beside the parental cell line, and two KO clones with specifically targeted IDH1^wt^/IDH1^mut^ alleles. Although the wild type allele is also targeted, the level of its product, α-KG, was unaffected, which suggests that glutaminolysis and/or other IDH isomers could have contributed to its activity [36]. These results rule out a possible contribution of IDH1^wt^ loss to the observed anti-tumor effects in our knockout model. This concept is further supported by the inability of IDH1^wt^ overexpression to alter the anti-tumor phenotypes of the JJ012 and HT1080 IDH1 KO cells, as well as by the lack of any significant effects of IDH1 knockout on the properties of non-transformed C28 chondrocytes. Thus, we have successfully established a model system for IDH1^mut^ knockout chondrosarcoma cells.

IDH1/2 mutations have been frequently observed in gliomas, leukemia and cartilaginous tumors (6). The high mutation frequency suggests a causal rather than a bystander role of IDH mutations in tumorigenesis. To elucidate the function of IDH mutations in chondrosarcoma, we first evaluated the effects of IDH mutation on the oncogenic phenotypes of chondrosarcoma cells. We found that knockout of IDH1^mut^ resulted in a nearly complete D-2HG clearance and attenuated cell adhesion and migration as well as anchorage-independent growth without affecting cell proliferation. This is consistent with our previous observations using an IDH1^mut^ inhibitor that compromised D-2HG production, colony formation, and cell migration [9]. These promising in vitro results encouraged us to further understand the role of IDH1 mutation in tumor maintenance. We found that tumorigenic growth of both cell lines in nude mice was dramatically impaired upon IDH1^mut^ knockout. This is supported by studies from Rohle et al. and Ma et al. who respectively reported that knockdown of IDH1^mut^ by shRNA [37] and knockout of IDH1^mut^ by TALEN technology [38] similarly impaired the growth of HT1080 cells in mice, though the off-target effects were a concern in Ma’s study where only one KO cell line was examined. These in vitro and in vivo phenomena support the observation that IDH mutation is required for maintaining the tumorigenicity of IDH-mutant chondrosarcoma cells.

As molecular details are being unraveled, the emerging concept is that accumulation of the oncometabolite, D-2HG in IDH-mutant tumors results in inhibition of α-KG-dependent dioxygenases involved in DNA and histone demethylation through direct competition with α-KG. The resultant hypermethylation affects gene expression and cellular differentiation, thus promoting tumor formation [5,39]. However, the role for these epigenetic changes in chondrosarcomas is controversial. Suijker et al. have reported that CpG island methylation and histone H3K4, -9, and -27 trimethylation levels remained unchanged upon specific inhibition of IDH1^mut^ in chondrosarcoma cells [16]. On the other hand, Nakagawa et al. recently demonstrated that selective inhibition of IDH1^mut^ significantly reduced the levels of H3K4, -9 trimethylation and promotes normal chondrocyte differentiation [40]. This discrepancy suggests that IDH1^mut^ may promote chondrosarcoma growth through some mechanisms beyond the well-characterized epigenetic effects in other malignancies. In our study, we further pursued the underlying molecular mechanism by which IDH mutation promotes chondrosarcoma formation. We uncovered a novel role for IDH mutation in integrin activation inferred from the marked downregulation of integrin genes upon IDH1^mut^ knockout. Significantly, over half of integrin genes were downregulated in the dedifferentiated chondrosarcoma HT1080 cell line that formed tumors within one week and grew at a rapid rate in our xenograft model. Integrins are cell-surface receptors that mediate the interaction of cells with the microenvironment through binding to their ECM ligands. Upon ligand binding, activated integrins directly and indirectly initiate cellular signaling to regulate the biological behavior of the cells such as cell adhesion, proliferation, migration, invasion, and survival [41]. Integrin levels are frequently elevated in other aggressive tumor types [35], and activation of integrins has been implicated in many pathological processes including carcinogenesis, cell growth, angiogenesis, and metastasis [42,43]. Therefore, the decrease in expression of integrin genes could explain our prior findings. It is likely that loss of integrin expression induced by IDH1^mut^ inactivation compromises the interactions between the chondrosarcoma cells and the ECM, hence the attenuated in vitro and in vivo tumorigenicity of the IDH1^mut^ KO cells. Although more studies are needed to unveil the role of integrins in cell survival, we believe that aberrant integrin signaling upon IDH1^mut^ knockout compromised the survival ability of the inoculated tumor cells which may be responsible for the slower tumor growth in the IDH1^mut^ KO groups. Moreover, in our study, we identified that integrins α5 and β5 along with their heterodimer complexes α5β1 and αvβ5 were closely associated with IDH1^mut^. To examine our hypothesis, we further investigated the role of these two integrin complexes in chondrosarcoma cells. As expected, they proved to contribute to cell migration and/or adhesion. It should be noted that cell adhesion to vitronectin was not altered in the HT1080 KO clones, or in HT1080 cells pretreated with neutralizing ITGαvβ5 antibody. These data indicate that cell adhesion to vitronectin is not affected by IDH1^mut^ knockout and this integrin complex does not play a critical role in HT1080 cell adhesion. However, this result does not exclude a critical role of other IDH mutation associated integrins in cell adhesion to different extracellular matrices. Thus, modulation of integrins by IDH mutation appears to be cell type specific. Furthermore, since integrins are known to have an important role in angiogenesis [25], we asked whether downregulation of integrins altered the angiogenic pathway in chondrosarcoma tumors. Indeed, as indicated by the expression of endothelial marker, ERG, we found that vascular density was significantly lower in the JJ012 IDH1^mut^ KO tumors compared to the parental control tumors, which could also be responsible for the slower growth of these tumors. Nonetheless, our data support the notion that IDH mutations promote chondrosarcoma growth through modulation of integrins. The decrease in FAK phosphorylation in both JJ012 and HT1080 IDH1^mut^ KO cells compared with their parental controls, and the increase in FAK activation in these KO cells upon treatment with octyl-D-2HG, further suggest that the oncometabolite, D-2HG is involved in integrin modulation. A proposed model of mutant IDH1 function in human chondrosarcomas is illustrated in Figure 7.

As mediators of a wide spectrum of cancer cell activities, integrins represent an appealing target for cancer therapy. Integrin antagonists, such as the function-blocking monoclonal antibody against ITGα5β1, volociximab, and the ITGαvβ3 and αvβ5 inhibitor, cilengitide, have shown promising activity in preclinical and clinical studies [35,44,45], with relatively few side effects. Nonetheless, successful exploitation of anti-integrin therapeutics for cancer patients has proven challenging. Cilengitide, the best-studied integrin antagonist, failed to show clinical benefits in clinical trials on glioblastoma and prostate cancer [46,47]. This highlights the importance of continued research to determine the role of integrins in tumor progression and to identify the factors that could increase the effectiveness of these inhibitors. In this study, we demonstrated critical roles for integrins which make these molecules attractive targets in IDH-mutant chondrosarcomas. Considering the substantial downregulation of integrin genes, we believe that more integrin receptors in addition to α5 and β5, could be promising targets as their functions are further elucidated. It should be noted that several integrin genes such as ITGA7 in JJ012 IDH1^mut^ KO clones and ITGAX in HT1080 IDH1^mut^ KO clones were upregulated. This may be a result of tumor cell compensation through feedback networks triggered by the loss of other integrins. Therefore, targeting these additional integrins may also be crucial in order to abrogate all integrin signaling in tumors cells. Moreover, a series of inhibitors of IDH1^mut^, IDH2^mut^, or both, developed and evaluated in preclinical and clinical studies as single agents and in combination with other anticancer agents, have shown promising clinical benefits [19]. While the inhibitors, AG-120 and AG-221 elicit partial responses or complete remission in AML patients, preclinical responses in solid tumors are less consistent and less robust. To enhance the antitumor effect of IDH^mut^ inhibitors, particularly in low-grade gliomas, a few preclinical studies investigated synthetic lethal interactions of these inhibitors, with other reagents such as NAMPT inhibitors [48], Bcl-2 inhibitors [49], NRF2 inhibitors [50], PARP inhibitors [51], and tyrosine kinase inhibitors [52]. The combinatorial regimens have proven to be more effective. In our study, we linked integrins to IDH mutation, making the integrin molecules appealing candidates for combinatorial regimens for IDH-mutant chondrosarcomas. Our findings provide novel targets for chondrosarcoma patients who have no other treatment options. Further studies on the mechanism by which IDH mutation drives integrin expression and understanding the functions of IDH1^mut^-upregulated integrins may help identify additional novel targets for treating patients with IDH1-mutant chondrosarcomas.

## 5. Conclusions

In this study, we found that IDH1^mut^ knockout significantly suppressed the tumorigenicity of chondrosarcoma cells, and resulted in marked attenuation of chondrosarcoma formation and elimination of the oncometabolite, D-2HG production. We further demonstrated that loss of IDH1^mut^ led to widespread downregulation of integrin genes and deregulation of integrin-mediated processes. Our findings indicate that IDH1^mut^ knockout abrogates chondrosarcoma genesis through modulation of integrins. This observation suggests that integrin molecules are appealing candidates for combinatorial regimens with mutant IDH inhibitors for IDH-mutant chondrosarcomas.

## Figures and Tables

**Figure 1 cancers-12-00141-f001:**
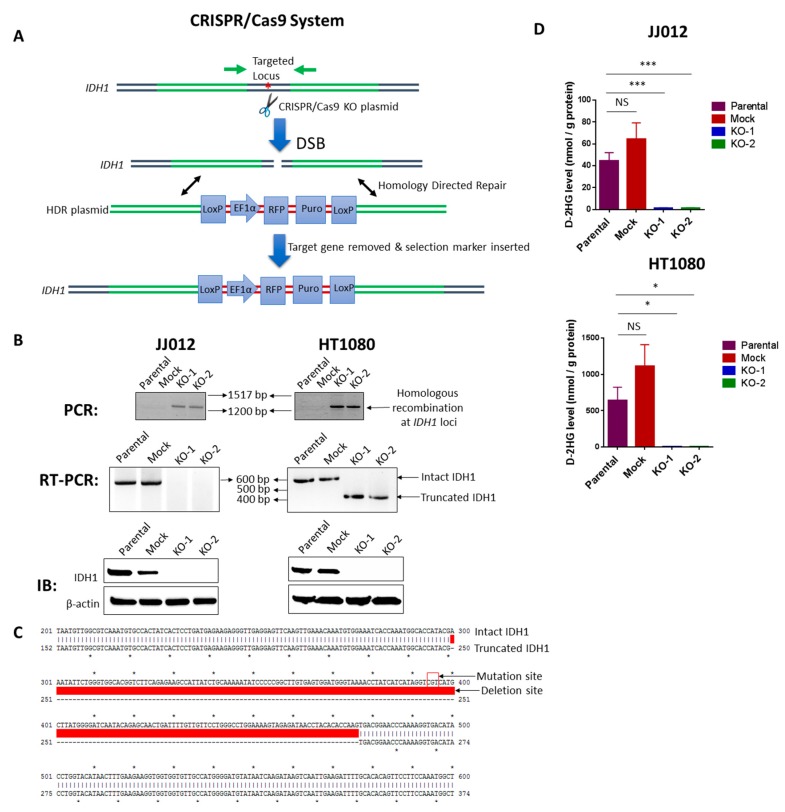
Knockout of IDH1^mut^ in two human chondrosarcoma cell lines. (**A**) Diagram of the CRISPR/Cas9 system. Both IDH1 CRISPR/Cas9 KO plasmid and HDR plasmid products consist of a pool of 3 plasmids. Each KO plasmid encodes a unique 20 nt sequence of gRNA which binds to target locus of *IDH1*. Cas9 cleaves the 5′ exons of *IDH1* targeted by the three gRNA plasmids at three specific sites and results in a disruption of *IDH1*. DNA containing DSB can be repaired by homology-directed repair (HDR) pathway. HDR plasmids feature two approximately 800 bp homology arms designed to specifically recombine with the DNA sequence surrounding the DSB and thus serve as a specific DNA repair template. When co-transfected with the corresponding KO plasmid, the HDR plasmid incorporates the RFP and puromycin resistance gene for selection of cells where Cas9-induced DNA cleavage has occurred. Red asterisk represents *IDH1* point mutation. (**B**) upper panel: PCR shows the homologous recombination at the *IDH1* loci in the KO clones of both cell lines; middle panel: RT-PCR shows loss of intact IDH1 transcripts in the KO clones of both cell lines; lower panel: immunoblot shows depletion of IDH1 protein in the KO clones of both cell lines. (**C**) Sanger sequencing and sequence alignment using ApE plasmid editor indicate a deletion of the IDH1 mutation site in HT1080-derived KO clones. Upper and lower arrows indicate the mutation site of intact *IDH1* and partial deletion site of truncated *IDH1*, respectively. (**D**) HPLC-MS analysis indicates that knockout of IDH1^mut^ dramatically reduced D-2HG production in both cell lines. Data represent mean ± SEM of at least five replicate samples. * *p* < 0.05, *** *p* < 0.001.

**Figure 2 cancers-12-00141-f002:**
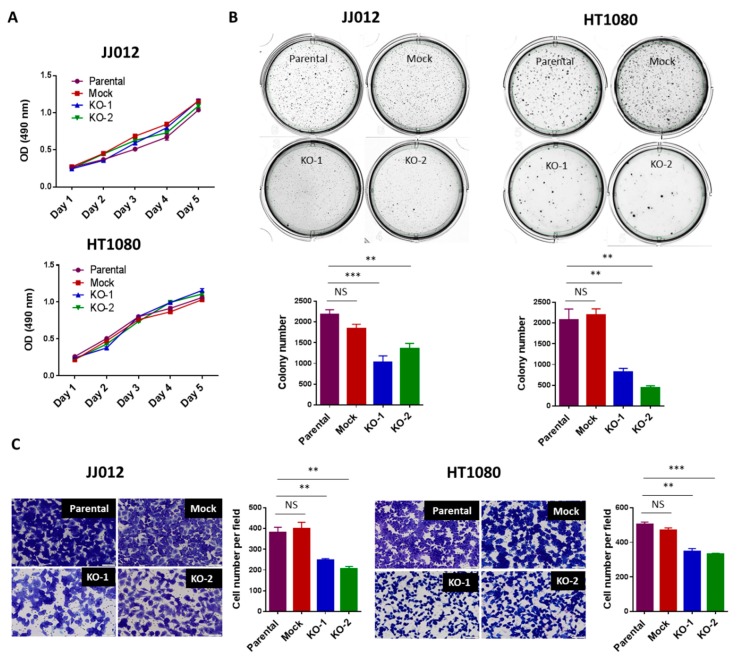
Loss of IDH1^mut^ attenuated the tumorigenicity of chondrosarcoma cells without affecting cell proliferation. (**A**) Loss of IDH1^mut^ did not alter cell proliferation. 2 × 10^3^ cells per well were cultured in 96-well plates for 1–5 days. Cells were incubated with MTS solution for 1 h and plate was read with a microplate spectrometer using a 490 nm filter. (**B**) Loss of IDH1^mut^ reduced soft-agar colonies. Soft-agar colony formation assay was performed. 5 × 10^3^ cells per well were seeded in 6-well plates and incubated for 10–14 days. Data indicates mean ± SEM of triplicate cultures and are representative of 3 independent experiments. (**C**) Loss of IDH1^mut^ reduced the migratory cell numbers. In vitro migration assay was performed using transwell chambers. 2.5 × 10^4^ cells per 24-well chamber or 2.5 × 10^5^ cell per 6-well chamber were seeded and incubated for 22h. Cells were then stained and counted manually after photographing the membrane through the microscope. Data are shown as mean ± SEM of triplicate cultures and are representative of three independent experiments. Original magnification, ×20 (scale bars: 100 μm). ** *p* < 0.01, *** *p* < 0.001.

**Figure 3 cancers-12-00141-f003:**
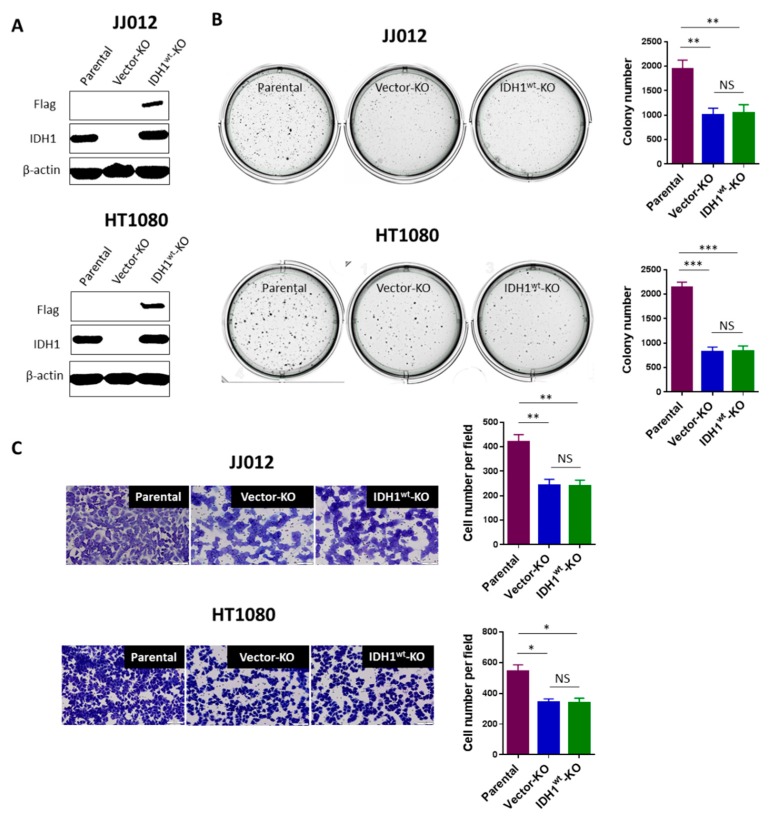
Loss of IDH1^wt^ is dispensable for the attenuated tumorigenicity of chondrosarcoma cells. (**A**) Immunoblot shows high levels of exogenous IDH1 (Flag) in the IDH1 KO cells from both cell lines. (**B**,**C**) Re-expression of IDH1^wt^ did not alter the decrease in colony formation or cell migration of the IDH1 KO cells. Histograms show mean ± SEM of triplicate cultures and are representative of 3 independent experiments. Original magnification, ×20 (scale bars: 100 μm). * *p* < 0.05, ** *p* < 0.01, *** *p* < 0.001.

**Figure 4 cancers-12-00141-f004:**
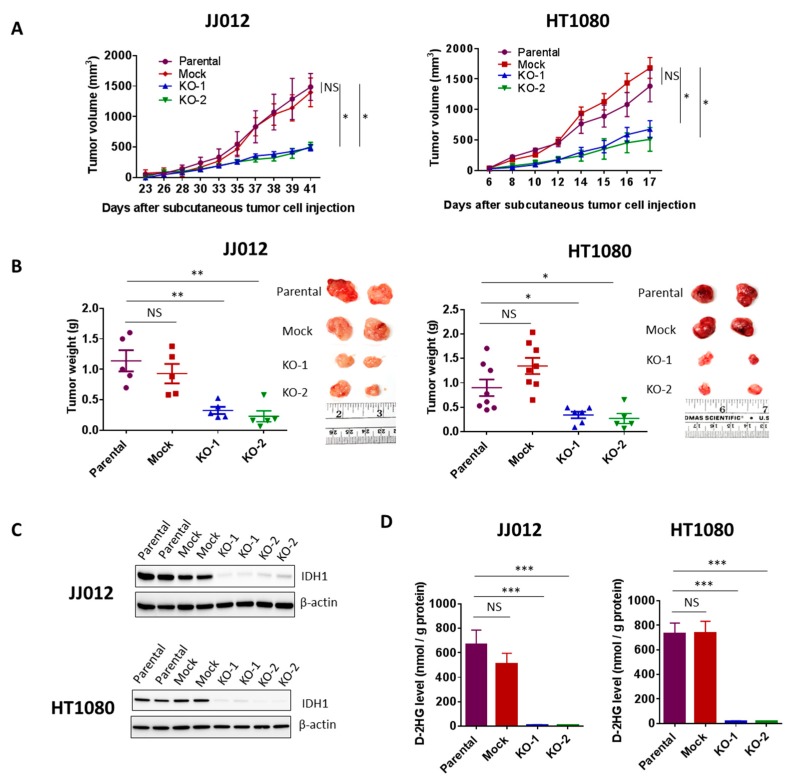
Loss of IDH1^mut^ led to suppression of chondrosarcoma formation and D-2HG production. (**A**) Effect of IDH1^mut^ knockout on tumor growth in subcutaneous chondrosarcoma xenograft models. 2 × 10^6^ cells from each group of JJ012 or HT1080 cell were injected subcutaneously into the flanks of nude mice (*n* = 8). Tumor volume was measured frequently with calipers once the tumors developed. (**B**) Tumor endpoint weight and photographs of two representative tumors from each group. Tumor weight was determined at 41 days post-injection of JJ012 groups and 17 days post-injection of HT1080 groups. Data represent average values with SEM in each group. (**C**) Immunoblots showing depletion of IDH1 in total protein lysates derived from IDH1^mut^ KO tumors. Two representative tumors per group are shown. (**D**) HPLC-MS analysis shows depletion of D-2HG production in the IDH1^mut^ KO clones-derived xenografts. Data represent mean ± SEM of at least five tumor samples from each group. * *p* < 0.05, ** *p* < 0.01, *** *p* < 0.001.

**Figure 5 cancers-12-00141-f005:**
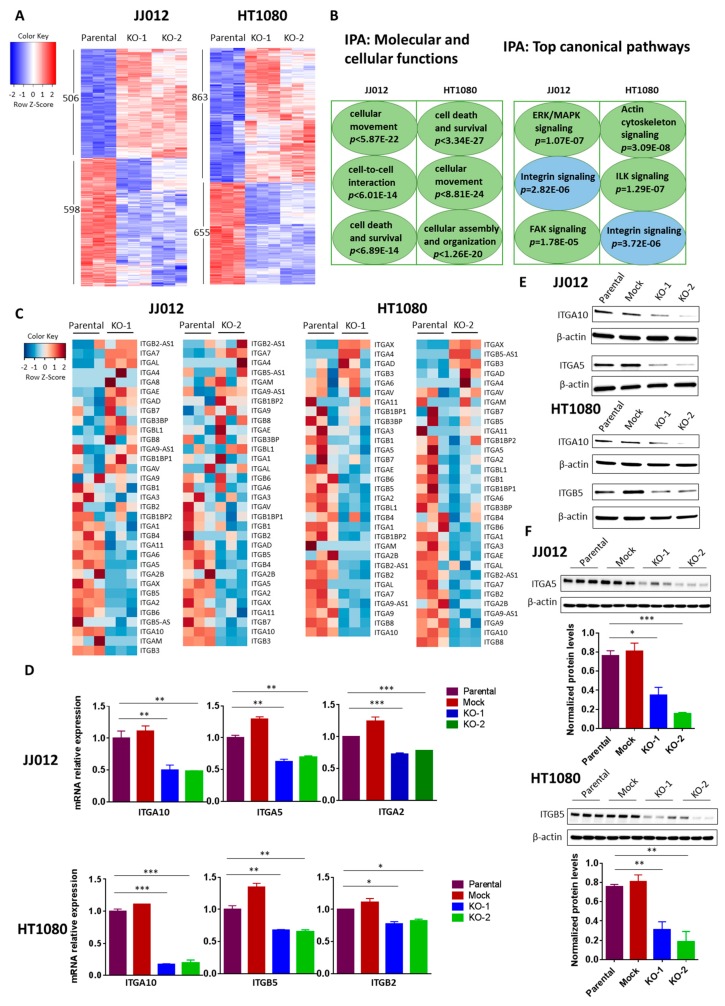
Effects of IDH1^mut^ knockout on gene expression in chondrosarcoma cell lines. (**A**) Heatmaps of differentially expressed genes in two IDH1^mut^ KO clones compared to parental control of each cell line. RNA-Seq analysis was performed in triplicate. Upon IDH1^mut^ knockout, the expression of 1104 genes and 1518 genes changed significantly in JJ012 and HT1080 cells, respectively (FDR-adjusted *p* value < 0.05). (**B**) IPA was performed with RNA-Seq data obtained in (**A**). (**C**) Heatmaps show the expression profiles of integrin genes upon IDH1^mut^ knockout. (**D**) Expression of ITGA10, ITGA5 and ITGA2 in the indicated groups of JJ012 cells; ITGA10, ITGB5 and ITGB2 in the indicated groups of HT1080 cells was quantified by qRT-PCR. The amount of transcript was normalized to GAPDH and the results are shown as fold-change relative to the parental control. Data are shown as mean ± SEM of triplicate values and are representative of three independent experiments. (**E**) Immunoblots of total protein lysates from control and IDH1^mut^ KO cells, probed with the indicated anti-integrin antibodies. (**F**) Immunoblots of total protein lysates from control and IDH1^mut^ KO tumors, probed with the indicated anti-integrin antibodies. Quantitation of signal in each group is normalized to β-actin. Three representative tumors per group are shown. * *p* < 0.05, ** *p* < 0.01, *** *p* < 0.001.

**Figure 6 cancers-12-00141-f006:**
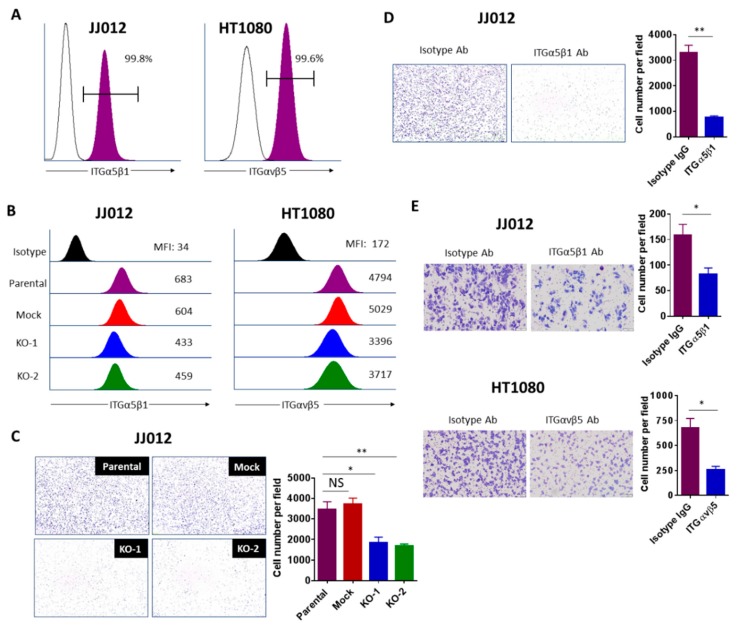
Integrins contribute to the tumorigenicity of chondrosarcoma cells. (**A**) Flow cytometry verifying high expression of ITGα5β1 and αvβ5 on the surface of JJ012 and HT1080 cells, respectively. Representative FACS data from 3 independent experiments are shown. (**B**) Representative flow cytometry histogram demonstrating downregulated expression of ITGα5β1 and αvβ5 in JJ012 and HT1080 IDH1^mut^ KO cells, respectively. Representative FACS data from three independent experiments are shown. (**C**) Cell adhesion assay of indicated JJ012 groups and (**D**) Cell adhesion assay of JJ012 cells pretreated with 1 μg/mL ITGα5β1-blocking antibody or its corresponding isotype control antibody. Cells were incubated on fibronectin-coated plates for 1–2h followed by washing, fixing, and staining. Cell adhesion was evaluated by counting the average number of attached cells per field. Results are shown as the mean ± SEM of triplicate cultures and are representative of three independent experiments. Original magnification, 10×. (**E**) Transwell migration assay of cells pretreated with 1 μg/mL ITGα5β1 (JJ012) or 10 μg/mL ITGαvβ5 (HT1080) blocking antibodies and their corresponding isotype controls for 1–2 h. The migrated cells were counted and the average cell number per field was determined. Data indicate mean ± SEM of triplicate cultures and are representative of three independent experiments. Original magnification, 20× (scale bars: 100 μm). * *p* < 0.05, ** *p* < 0.01.

**Figure 7 cancers-12-00141-f007:**
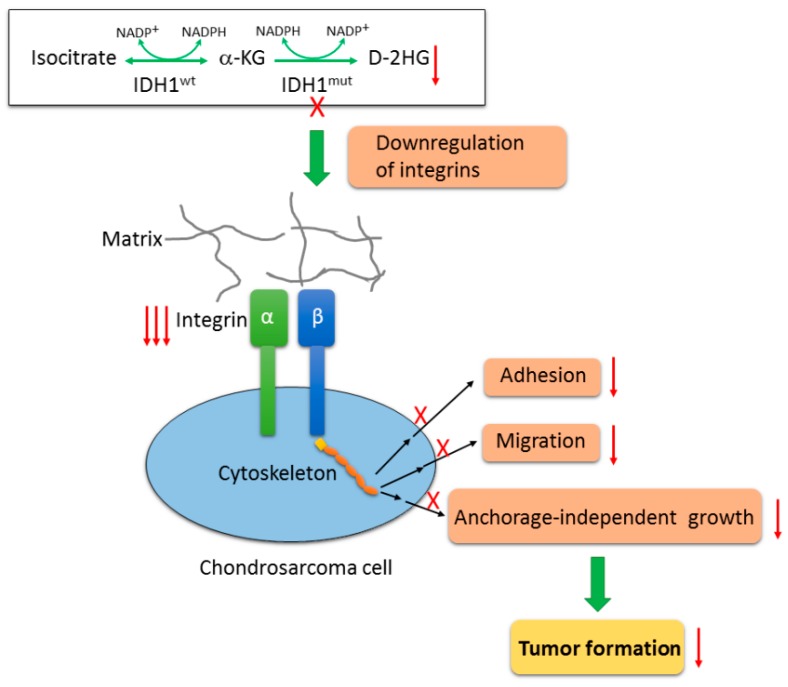
Proposed model of mutant IDH1 function in chondrosarcoma cells. Wildtype IDH enzymes catalyze the oxidative decarboxylation of isocitrate to form α-KG. IDH mutation confers on the mutant proteins a neomorphic activity by reducing α-KG into an oncometabolite, D-2HG. Inactivation of IDH1^mut^ by CRISPR/Cas9 technique nearly depleted D-2HG production, and subsequently attenuated the tumorigenicity of chondrosarcoma cells such as cell adhesion, cell migration and anchorage-independent growth, and impaired tumorigenic growth in vivo. This is achieved at least partially by the interruption of integrin-mediated cell-matrix interactions as a result of a wide downregulation of integrin genes upon loss of IDH1^mut^.

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
