# Peer review of "Mutant IDH1 Depletion Downregulates Integrins and Impairs Chondrosarcoma Growth"

_cancers, 2020, doi:10.3390/cancers12010141_

Round 1

Reviewer 1 Report

In the current manuscript by Li et al., the experiments were carefully done and the manuscript contains interesting observations.  Nevertheless, I have a concern that needs to be addressed before publication in Cancers.

Although the knockout of mutant IDH1could inhibit the tumor promotion in chondrosarcomas, it could not show suicidal effect on tumor cells. Are there possibilities that the knockout of mutant IDH1 induce the dormancy and/or resistance against anti-cancer drugs of tumor cells?

Reviewer 2 Report

Overall Considerations: Li et al described in their manuscript a new signaling into IDH1mut in chondrosarcoma. They identified for the first time a link between IDH1mut and integrins expression. This paper includes in vivo and in vitro data including RNAseq data. The paper is well written and data are convincing but some concerns need to be addressed.

Major points:

The first major point of this paper is that data do not explain the proliferation observed in vivo. Modulation of integrins signaling suggest that cells will be more metastatic not more proliferative. Do the authors see metastatic processes in vivo? (In lungs for example?) Since both integrins can impact angiogenesis, do the authors see any differences in vascularization, VEGF expression in vivo…?

The second major point is that this paper does not described the mechanism of D-2HG on integrin expression and signaling. Is adding D-2HG in the medium modulates both integrins signaling (FAK, PAK4 activation…)(by confocal), modulates cells migration/invasion? That will at least demonstrate that integrins over-expression and signaling is dependent of D-2HG, even though that will not explain the mechanism. Integrin alpha 10 is also strongly decreases in the KO, why the authors did not look at this integrin that can bind collagen, since we are talking chondrosarcoma? Do the authors observed a positive correlation between IDH mut and integrins Alpha 5 and beta5 in cohorts by meta-analysis?

Fig.2: The authors referred to their previous published data two times in PLoSOne 2015 questioning the novelty of the results. Supplemental data?

Fig.3: positive control with a-KG production should be shown with IDH1 wt

The fact that the authors used KO for IDH1mut KO and KO for IDH1wt is very confusing.

Minor:

L432 “heterodimers”

Reviewer 3 Report

This manuscript describes the effects of altering IDH1 in chrondrosarcoma cells via genetic deletion of IDH1 by the CRISPR/Cas9 system. The effects of the genetic manipulation confirms previously published data by the authors obtained via pharmacological inhibition of mutant IDH1 which leads to alterations in chondrosarcoma cells’ abilities to grow in anchorage independent conditions and migration as well as decrease in tumor growth in vivo. RNA-seq analysis lead to the identification of integrins as a potential mechanism for the oncogenic action of mutant IDH1 which was confirmed via flow cytometry and adhesion assays.

The strengths of this manuscript are the use of two cell different cell lines in this study, including in vitro  and in vivo experiments, verification of specificity of mutant effects vs removal of wild type gene effects. The weakness is lack of use of the previously described pharmacological interventions to link previously reported data to the potential integrin-related mechanism.

I have the following comments:

Line 286 in the results section has a typographical error that significantly alters the meaning of the sentence. The sentence should state “Moreover, anchorage-independent cell growth is the ability of transformed cells to grow independently OF (not on) a solid cell surface.”

Figure 5:

In figure 5 please define what IPA means, the assumption is ingenuity pathway analysis but this should be stated.

Western blots should have densitometric measurements along with error bars and statistics.

Discussion:

It would be beneficial to repeat some of this work with the pharmacological inhibitor to test if pharmacological blockage can affect integrin expression and function in these cells.

It would be beneficial for the authors to replicate some of the integrin data with the IDH1 pharmacological inhibitor that was utilized in their previous study.

The authors should include the vitronection adhesion data into Figure 6 instead of just mentioning it in the discussion as dns. The authors should comment more on why there were not alterations in binding to vitronection by HT1080 cell lines and suggest some alternatives.

Reviewer 4 Report

The current work is interesting but not technically sound. There are several major concerns:

No normal cells have been used in the current study. In methods and material write separately for each assay in detailed fashion (Cell proliferation, colony formation, transwell migration and cell adhesion assays). Migration assay should be performed using wound healing along with invasion assay. Integrin expression profile should be provided using flow cytometry. In the in vivo experiments provide images with all tumors, as the authors have used 8 mice per group. The image should be taken with all tumors kept in one plane together with the scale. In Figure 4B, it seems that the tumors and the scale has been set up separately.

Round 2

Reviewer 2 Report

Major points:

The authors re-expressed IDH in KO mice in Fig.3. Even though, we can clearly see an increased in IDH (Fig.3A), they observed an increased in a-KG, no impact on cell migration and number of colonies suggesting that the IDH overexpressed construct is not functioning as expected. What is the level of D2HG that should increase and consequently a-KG should decrease. Since the authors did not bring any controls attesting the functionality of the over expressed IDH, they cannot conclude the content presented in Fig.3 and therefore these data are irrelevant and question the role of IDH KO described in the paper.

The authors wrote in their response to the reviewer that they observed an impact on angiogenesis (VEGF). This data should be included in the manuscript to give at least one signaling that may be altered by IDH KO that can explain the phenotype observed in vivo on tumor progression (volume and weight) in Fig.4.

Concerning the comment on FAK, the approach by adding Octyl-D-2HG can be considered as a first approach. Now what about (as it was asking in the review) FAK activation in cells KO versus CT only? The authors shouldn’t need a treatment to see if FAK activation is modulated. If WB is not enough sensitive, Immunofluorescence and Imaging by simple confocal microscopy should be considered.

Reviewer 4 Report

Comments have been addressed in the amended version.

Author Response

We greatly appreciate your recognition of our work.

This manuscript is a resubmission of an earlier submission. The following is a list of the peer review reports and author responses from that submission.